# Straightforward construction of functionalized *γ*-lactams via conjugated-engineered covalent organic framework photocatalysed cascade reactions

Xiangfeng Lin[1,7], Jianguo Li[1,7], Jiaxian Zheng[1,7], Xiaowei Cai[1], Liwei Wang ⓘ[2] ✉, Rongjian Sa ⓘ[2] ✉, Chuanling Si ⓘ[3] ✉, Dong Jiang ⓘ[4,5] ✉, Yunqing Kang ⓘ[4], Jie Wang[5], Yusuke Yamauchi ⓘ[4,5,6] & Zhanhui Yuan ⓘ[1] ✉

Straightforward synthetic approach for direct constructing functionalized *γ*-lactams is highly valuable given their ubiquity in bioactive molecules yet challenging due to instability of the products and difficulties in controlling diastereoselectivity. We herein report the design and synthesis of a naphthyl-based pyrene-containing covalent organic framework and utilize it as heterogeneous photocatalyst in visible-light catalytic cascade reactions between tryptamine-derived isocyanides and phosphine oxides. A broad range of 3-(2-aminophenyl)-*γ*-lactams are isolated in good yields under mild conditions and 3-(2-isocyanobenzyl)-indoles are also tolerated, giving (2-aminophenyl)-tetra-hydroquinolines in moderate yield. Experimental studies and theoretical calculations reveal that a proton-coupled electron transfer process occurs between the excited covalent organic framework and phosphine oxide, which subsequently triggers a series of cascade reactions involving cyclization, semipinacol rearrangement, and dehydrogenation/hydration processes. Notably, the in situ generated spiroindolenine is identified to be the key intermediate. Illustrated by the efficient synthesis of 3-(2-aminophenyl)-*γ*-lactams, this work sets a precedent for the development of heterogeneous photocatalytic strategies for constructing complex bioactive molecules.

Cyclic amides are core pharmacophores found across natural products and clinically used drugs. *β*-lactams anchor the antibiotic arsenal[1,2], whereas the five-membered *γ*-lactam ring continues to serve as the critical scaffold in marketed agents for cancer, depression, and other diseases[3–16]. As an essential structural framework of functionalized *γ*-lactams, the 3-(2-aminophenyl)-*γ*-lactams are known for their pronounced biological activity (Fig. 1a)[17–19]. Several examples involving the synthesis of *α*-OH substituted (2-aminophenyl)-*γ*-lactams such as chimonamidine and donaxaridine, have been developed through a multi-step functionalization process of 3-hydroxy-2-oxindoles[20–24]. Ren et al.

[1]College of Materials Engineering, Fujian Agriculture and Forestry University, Fuzhou, P. R. China. [2]College of Materials and Chemical Engineering, Minjiang University, Fuzhou, P. R. China. [3]State Key Laboratory of Biobased Fiber Materials, Tianjin Key Laboratory of Pulp and Paper, Tianjin University of Science and Technology, Tianjin, P. R. China. [4]Department of Materials Process Engineering, Graduate School of Engineering, Nagoya University, Nagoya, Japan. [5]Australian Institute for Bioengineering and Nanotechnology (AIBN), The University of Queensland, Brisbane, QLD, Australia. [6]Department of Chemical and Biomolecular Engineering, Yonsei University, Seoul, South Korea. [7]These authors contributed equally: Xiangfeng Lin, Jianguo Li, Jiaxian Zheng. ✉e-mail: wlw@mju.edu.cn; rjsa@mju.edu.cn; sichli@tust.edu.cn; dongjiang@toki.waseda.jp; zhanhuiyuan@fafu.edu.cn

**Fig. 1 | The synthesis of 3-(2-aminophenyl)-γ-lactams via conjugated-engineered covalent organic framework photocatalysed cascade reactions.** **a** 3-(2-Aminophenyl)-γ-lactams and several relevant natural products. **b** The synthesis of spiroindolenines via photoinduced cyclizations and designed plan involving the synthesis of 3-(2-aminophenyl)-γ-lactams via photocatalytic cascade reactions. **c** This work: The synthesis of 3-(2-aminophenyl)-γ-lactams and derivatives via NPy-DMTP-COF photocatalytic cascade reactions.

developed a copper/iodine co-catalyzed oxygenation transannulation, providing direct access to the donaxaridine and its derivatives[25]. Owing to the intrinsic lability of the target intermediates, extra functional groups must be pre-embedded in the substrates, a requirement that inevitably compromises both atom economy and overall cost efficiency[21]. Therefore, the development of a sustainable and straightforward synthetic approach for synthesizing this subclass of functionalized $\gamma$-lactams from widely available starting materials would provide significant insights and inspiration in this domain.

Our initial proposal for constructing 3-(2-aminophenyl)-$\gamma$-lactams was transformation from spiroindolenines, which can be formed via a photocatalytic cascade reaction between tryptamine-derived isocyanide and nucleophilic radical via dearomatization reactions[26–34]. Tryptamine-derived isocyanides, featured an indole core with an isocyanide chain at the C3 position, have exhibited exceptional efficacy in various chemical reactions. Diverging from conventional catalytic methods, Orru et al. innovated a catalyst-free dearomatization under visible light that yields a series of spiroindolenines **A** (Fig. 1b, black solid arrow)[35]. The crux of this process is primarily attributed to the formation of an imidoyl radical intermediate, arising from the interaction between phenacyl radical and tryptamine-derived isocyanide. Despite these advances, the development of a photocatalytic cascade reaction between tryptamine-derived isocyanide and radical precursor initiated by a single electron transfer (SET) oxidation or PCET process involving reductive quenching cycle, such as the reaction with in-situ generated phosphoryl radical from phosphine oxides, remains a significant challenge (Fig. 1b, blue dashed-line box). Notably, Yu et al. have developed a photocatalytic synthesis of phosphorylated N-heteroarenes from aryl isocyanides and phosphine oxides via a 4CzIPN-catalyzed PCET mechanism[36]. Wu et al. achieved similar outcomes using cobaloxime photocatalysis with $H_2$ as byproduct[37]. Building on existing examples of photocatalytic cascade reactions involving aryl isocyanides[38–44], we envision that spiroindolenine **B** could be constructed via a photocatalytic cascade reaction between the tryptamine-derived isocyanide and phosphine oxide (Fig. 1b, black dashed arrow). Subsequently, **Int-B** could be generated through hydration of spiroindolenine **B**, followed by a semipinacol rearrangement, which would be a powerful method for constructing ketones with quaternary centers and can occur in both acidic[45–62] and basic[63] environments. Ultimately, the **Int-B** would undergo a dehydrogenation/hydration process, forming the (2-aminophenyl)-$\gamma$-lactams. Realizing the above transformation critically depends on the development of an efficient photocatalyst.

Covalent organic frameworks (COFs) are porous crystalline materials composed of organic building blocks, known for their tunable porosity, extended π-conjugation, adjustable bandgaps, high surface areas and stability. These properties make COFs highly attractive for photocatalysis[64–75]. Recently, two-dimensional COFs (2D-COFs) have shown exceptional performance as heterogeneous photocatalysts, driving various chemical transformations under light irradiation[76–81]. Among them, benzyl-based pyrene-containing COFs (Py-COFs) synthesized by the condensation between 4,4′,4″,4‴-(pyrene-1,3,6,8-tetrayl)tetraaniline (Py) and diarylglyoxal has demonstrated impressive photocatalytic activity in a range of organic reactions[82–87]. One of the key advantages of Py-COFs lies in their broad π-conjugated systems, which possess electron-withdrawing characteristics that readily enable the capture of electrons in PCET events. Moreover, the photocatalytic properties of pyrene-containing COFs can be precisely tuned by introducing aryl groups with diverse conjugated systems. In addition, Py-COFs have demonstrated remarkable efficacy as photocatalysts in reducing H(I) species to hydrogen gas[88–90]. It is believed that meticulously engineered pyrene-based COF catalysts, featuring optimized conjugation systems and pore structures, will significantly enhance the efficiency and sustainability of our designed cascade catalytic pathways.

Herein, we substituted the phenyl group with a naphthyl group at the 1,3,6,8 positions of pyrene and synthesized NPy-DMTP-COF. A series of heterogeneous photocatalytic cascade reactions between tryptamine-derived isocyanides and phosphine oxides mediated by NPy-DMTP-COF was established. This reaction affords a broad range of 3-(2-aminophenyl)-$\gamma$-lactams in moderate to high yields with excellent diastereoselectivities ( > 20: 1) (Fig. 1c). Specifically, 3-(2-isocyanobenzyl)-indoles are also tolerated, delivering a variety of (2-aminophenyl)-tetrahydroquinolines in moderate yield. Comprehensive mechanistic investigations indicate that the in-situ generated spiroindolenine serves as a pivotal intermediate in the cascade reaction, which encompasses cyclization, semipinacol rearrangement, hydration and dehydrogenation processes.

## Results and discussion
### Synthesis and characterizations of NPy-DMTP-COF
Initially, NPy-DMTP-COF was synthesized via a condensation reaction between 1, 3, 6, 8-tetrakis(2-aminonaphthyl)pyrene (NPy) and 2, 5-dimethoxyterephthalaldehyde (DMTP) (Fig. 2a and Supplementary Fig. 1). Compared to the reported Py-DMTP-COF (Fig. 2b), NPy-DMTP-COF (Fig. 2c) exhibits stronger electron-withdrawing characteristics, suggesting a greater ability to accept electrons during the PCET process. Additionally, we envisage that its more extensively conjugated π-system would enhance visible-light absorption, making NPy-DMTP-COF a promising photocatalyst. The PXRD pattern of NPy-DMTP-COF exhibits four prominent diffraction peaks, with the most-intensive one at 3.28° and the three other peaks at 5.12°, 6.60°, and 23.64°. These peaks are assigned to the (100), (110), (220), and (001) facets, respectively (Supplementary Fig. 2a). The simulated PXRD pattern of the eclipsed AA stacking conducted by Materials Studio Software is in good agreement with the experimental PXRD patterns with a P1 space group. The Pawley refinement cell parameters are a = 26.9 Å, b = 3.8 Å, c = 27.0 Å, $\alpha = 90°$, $\beta = 90°$ and $\gamma = 90°$, with good agreement factors $R_{wp} = 5.00\%$ and $R_p = 2.73\%$ for NPy-DMTP-COF (Supplementary Table 1). The NPy-DMTP-COF shows remarkable chemical and structural stability, as almost identical PXRD patterns are obtained after treatment with 14 M NaOH, triethylamine, conc. HCl, trifluoroacetic acid and boiling water (Supplementary Fig. 2b). Scanning electron microscopy (SEM) images reveal that NPy-DMTP-COF adopts a uniform rod-like morphology with an average diameter of approximately 65 nm (Supplementary Fig. 3). High resolution transmission electron microscopy (HR-TEM) images show distinct ordered fringes, confirming its high crystallinity and revealing a quadrilateral pore structure with a periodicity of approximately 2.6 nm and particle size of around 60 nm (Supplementary Fig. 4). Dynamic light scattering (DLS) measurements, performed on NPy-DMTP-COF dispersed in acetonitrile at room temperature, reveal a Z-average size of 41.05 nm and a polydispersity index (PDI) of 0.115 (Supplementary Fig. 5). The formation of imine linkage of NPy-DMTP-COF was confirmed by Fourier transform infrared (FT-IR) spectra. The disappearance of the N−H vibration at 1630, 3460 and 3370 cm⁻¹ and appearance of the C=N vibration at 1590 cm⁻¹ confirm the high condensation degree of NPy-DMTP-COF (Supplementary Fig. 6). The chemical structure was further elucidated using solid-state ¹³C cross-polarization/total sideband suppression (¹³C-CP/TOSS) spectroscopy. Chemical shifts at 152.4 ppm assigned to ¹³C signal of C=N bond indicate the successful formation of NPy-DMTP-COF with the imine linkage (Supplementary Fig. 7). Nitrogen adsorption-desorption isotherms demonstrate that the NPy-DMTP-COF has a mesoporous structure with a BET surface area of 1334 m²/g, a pore volume of 1.27 cm³/g and a pore width of 2.35 nm (Supplementary Fig. 8 and Supplementary Fig. 9), matching well with the simulated values and HR-TEM results. Thermogravimetric analysis (TGA) shows that NPy-DMTP-COF remains stable up to a temperature of 350 °C under both nitrogen and oxygen atmospheres, confirming its excellent thermal stability (Supplementary Fig. 10).

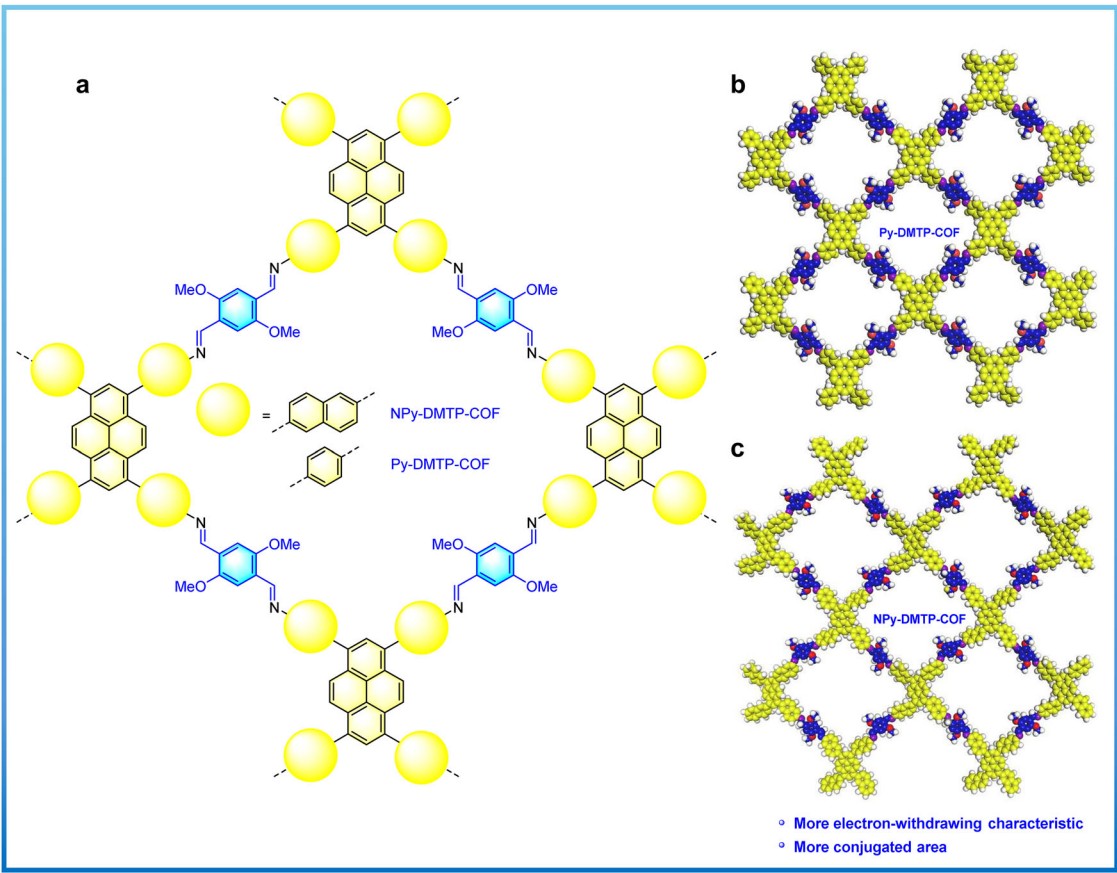

**Fig. 2 | Structures of Py-DMTP-COF and NPy-DMTP-COF. a** Chemical structures of Py-DMTP-COF and NPy-DMTP-COF. **b, c** Graphic view of one layer of Py-DMTP-COF and NPy-DMTP-COF simulated by Material Studio. (blue, yellow, C; purple, N; red, O; white, H).

## Reaction optimizations

Upon successfully synthesizing the NPy-DMTP-COF, we next evaluated its photocatalytic activity in the cascade reaction. The tryptamine-derived isocyanide **1a** was selected the as model substrate, along with 2.0 equivalents of diphenylphosphine oxide **2a** as the radical precursor and 2.0 equivalents of NaHCO₃ as the base. The reaction was carried out under a 40 W blue LED light source with a wavelength of 420 nm at room temperature (Table 1). The cascade reaction catalyzed by NPy-DMTP-COF yielded the $\gamma$-lactam **3a** as the predominant product with a satisfactory yield of 73% in CH₃CN at ambient temperature (Table 1, entry 1). Py-DMTP-COF, obtained through the condensation between 4,4′,4″,4‴-(pyrene-1,3,6,8-tetrayl) tetraaniline (Py) and 2, 5-dimethoxyterephthalaldehyde (DMTP) according to Yang's procedure[82] produced **3a** in a yield of only 23% (Table 1, entry 2). Several representative homogeneous photocatalysts displayed disappointing results (Table 1, entries 3–5). We speculate that the photocatalyst primarily facilitates the generation of phosphoryl radicals from compound **2a**. Diffuse-reflectance UV0-vis spectroscopy (UV-vis DRS) indicates that NPy-DMTP-COF and Py-DMTP-COF exhibit comparable absorption maxima around 420 nm (Supplementary Fig. 11). Therefore, the underlying rationale behind this disparity may be attributed to the redox potential of the photocatalysts. Based on an excitation energy ($E^{0-0}$) of 2.28 eV obtained through absorption and emission spectra of NPy-DMTP-COF (Supplementary Fig. 12 and Supplementary Table 2, the excitation and emission spectra of Py-DMTP-COF are shown in Supplementary Fig. 13), the excited-state redox potential of the PC*/PC⁻ couple is calculated to be 1.56 V. Cyclic voltammetry (CV) studies reveal that NPy-DMTP-COF possesses a redox potential of −0.72 V, stemming from the PC/PC⁻ redox couple (Supplementary Fig. 14, orange line).

Electrochemical measurements further reveal that compound **2a** exhibits an oxidation potential of 1.20 V (Supplementary Fig. 15). These results imply the potential for an electron transfer from **2a** to the excited-state COF (COF*), coupled with a proton transfer from **2a** to the base (HCO₃⁻), resulting in the formation of the phosphoryl radical and the COF radical anion COF⁻. The inefficacy of Py-DMTP-COF and representative homogeneous photocatalysts further supports this mechanistic rationale. For instance, *fac*-Ir(ppy)₃ and [Ir(ppy)₂(dtbbpy)]PF₆ exhibit excited-state redox potentials of only + 0.31 V (*vs*. SCE) and + 0.66 V (*vs*. SCE)[91], respectively, which are insufficient to oxidize **2a** ($E_{1/2}$(PC*/PC⁻) = + 1.20 V *vs*. SCE) to generate the phosphoryl radical. Similarly, Py-DMTP-COF shows a relatively low excited-state potential ($E_{1/2}$(PC*/PC⁻) = 1.04 V *vs*. SCE), accounting for its poor catalytic performance. In addition, 4CzIPN displays a reduction potential of −1.21 V[92], implying that its radical anion (4CzIPN⁻) is less prone to return to the ground state via electron transfer, further limiting its activity under these conditions. To evaluate the structural contribution of the extended COF, we tested the activity of the discrete molecular analog (NPy-MP) and the monomer (NPy). Neither exhibited noticeable catalytic activity, highlighting the importance of the ordered, porous framework in promoting the photocatalytic cascade reaction (Table 1, entries 6 and 7). Moreover, commercially available g-C₃N₄ and graphene oxide also failed to produce detectable amounts of **3a** (Table 1, entries 8 and 9), probably owing to their lack of well-defined porosity and extended π-conjugation.

The optical band gap of NPy-DMTP-COF is determined to be 2.12 eV through a Tauc plot analysis (Supplementary Fig. 16). Furthermore, electrochemical Mott-Schottky plots, combined with the optical band gaps, reveal that the conduction band (CB) and valence band (VB)

## Table 1 | Optimizations of the reaction

| Entry | PC | Base | Solvent | Yield/%[a] |
|---|---|---|---|---|
| 1 | NPy-DMTP-COF | NaHCO$_3$ | CH$_3$CN | 73 |
| 2[b] | Py-DMTP-COF | NaHCO$_3$ | CH$_3$CN | 23 |
| 3 | fac-Ir(ppy)$_3$ | NaHCO$_3$ | CH$_3$CN | 0 |
| 4 | [Ir(ppy)$_2$dtbbpy)]PF$_6$ | NaHCO$_3$ | CH$_3$CN | 0 |
| 5 | 4CzIPN | NaHCO$_3$ | CH$_3$CN | 0 |
| 6 | NPy | NaHCO$_3$ | CH$_3$CN | 0 |
| 7 | NPy-MP | NaHCO$_3$ | CH$_3$CN | 0 |
| 8 | g-C$_3$N$_4$ | NaHCO$_3$ | CH$_3$CN | 0 |
| 9 | Graphene oxide | NaHCO$_3$ | CH$_3$CN | 0 |
| 10 | NPy-DMTP-COF | Na$_2$HPO$_4$ | CH$_3$CN | 70 |
| 11 | NPy-DMTP-COF | KHCO$_3$ | CH$_3$CN | Trace |
| 12 | NPy-DMTP-COF | CsHCO$_3$ | CH$_3$CN | 0 |
| 13 | NPy-DMTP-COF | Na$_2$CO$_3$ | CH$_3$CN | 0 |
| 14 | NPy-DMTP-COF | NaHCO$_3$ | DCM | 0 |
| 15 | NPy-DMTP-COF | NaHCO$_3$ | Toluene | 0 |
| 16 | NPy-DMTP-COF | NaHCO$_3$ | THF | 21 |
| 17 | NPy-DMTP-COF | NaHCO$_3$ | DMF | Trace |
| 18 | NPy-DMTP-COF | - | CH$_3$CN | 0 |
| 19[b] | NPy-DMTP-COF | NaHCO$_3$ | CH$_3$CN | 43 |
| 20[c] | NPy-DMTP-COF | NaHCO$_3$ | CH$_3$CN | 0 |
| 21[d] | NPy-DMTP-COF | NaHCO$_3$ | CH$_3$CN | 0 |

All reactions were carried out on a 0.1 mmol scale with 1.0 eq. isocyanide **1a**, 2.0 eq. diphenyl-phosphine oxide **2a**, 2 mol% catalyst and 2.0 eq. base in 1.0 mL solvent under irradiation of blue LEDs with N$_2$ protection at room temperature. DCM is Dichloromethane. THF is tetrahydrofuran. DMF is N,N-dimethylformamide. 4CzIPN is commercial 2,4,5,6-tetrakis(carbazol-9-yl)-1,3-dicya-nobenzene. NPy [6,6',6'',6'''-(pyrene-1,3,6,8-tetrayl)tetrakis(naphthalen-2-amine)] and NPy-MP [(1E,1'E,1''E,1'''E)-N,N',N'',N'''-(pyrene-1,3,6,8-tetrayltetrakis(naphthalene-6,2-diyl))tetrakis(1-(2-methoxyphenyl)methanimine)] are both illustrated in the Supplementary Information. [a]Isolated yield. [b]The reaction was performed with 1.0 eq. base. [c]The reaction was performed in the dark. [d]The reaction was performed under air.

positions of NPy-DMTP-COF are well aligned to facilitate the PCET process (Supplementary Fig. 17 and Supplementary Fig. 18).

Subsequently, a series of bases were evaluated for their ability to facilitate the generation of the phosphoryl radical. Among them, only NaHCO$_3$ and Na$_2$HPO$_4$ produced the desired lactams with satisfactory results (Table 1, entry 1 vs entries 10-13). The solvent screening further revealed that the catalytic reactions proceeded efficiently in polar solvents, with acetonitrile affording the optimal yield (Table 1, entry 1 vs entries 14-17). The essential role of the base for the cascade reaction was confirmed by control experiments (Table 1, entry 18), and a significant reduction in yield was observed when using 1.0 equivalent of NaHCO$_3$ (Table 1, entry 19). These results indicate that the base might participate in both the cyclization and subsequent processes. As anticipated, no product was formed in the absence of light or under air atmosphere (Table 1, entries 20 and 21). Therefore, the optimized conditions involved the use of NPy-DMTP-COF as catalyst and NaHCO$_3$ as the base in CH$_3$CN at room temperature (Table 1, entry 1).

### Substrate scope
Under the optimized reaction conditions, we next explored the substrate scope with various phosphine oxides and tryptamine-derived isocyanides to investigate the generality of this photocatalytic cascade

reaction (Fig. 3). The protocol proved applicable to a broad range of diarylphosphine oxides bearing diverse substituents, including methyl, halogens, alkoxy groups and phenyl (**3a**–**3i**), affording the corresponding lactams in 53-83% yields. Both electron-donating and electron-withdrawing substituents at either the para- or meta-positions of the benzene ring were well tolerated. In addition, a 2-thienyl substituted phosphine oxide also participated smoothly, affording the desired lactam **3j** in 56% yield. Furthermore, phosphine oxides featuring two different aryl substituents demonstrated excellent reactivity, successfully yielding products **3k** and **3l** in good yields with d.r. values exceeding 20: 1. Notably, alkylphenylphosphine oxides were identified as suitable phosphorus reagents for this protocol, affording the corresponding products **3m**–**3o** in moderate yields while maintaining d.r. values exceeding 20:1. In contrast, dialkylphosphine oxides remained unreactive toward 1a, probably because the required PCET process between dialkylphosphine oxide and NPy-DMTP-COF* is disfavored (Supplementary Fig. 19). Single-crystal X-ray diffraction analysis of 3a verified the structure.

After establishing the scope of phosphine oxides, the effects of tryptamine-derived isocyanides on the reaction outcome were subsequently evaluated. As shown in Fig. 3, a wide range of tryptamine-derived isocyanides bearing different substituted aryl groups reacted smoothly with diarylphosphine oxides, affording the corresponding 3-(2-aminophenyl)-γ-lactams **3p**–**3x** in moderate to good yields (56-86%). In particular, 2-phenyl substituted tryptamine-derived isocyanide was well tolerated, affording the product **3y** in 70% yield.

The mild reaction conditions and high functional group tolerance could facilitate the late-stage functionalization of complicated drug or bioactive molecules. The reactions were highly compatible with tryptamine-derived isocyanides from Clofibrate, Fenofibrate and Cloquintocet-mexyl, giving the desired lactams **3z**–**3ab** in 61-82% yields. In addition, the 2-aryl substituted Celestolide-derived isocyanide was proved to be suitable for the reaction, affording the lactam **3ac** in 62% yield. Furthermore, acyloxy substituted isocyanides from Palmitic acid, Bezafibrate and Indomethacin were also well tolerated in the optimized conditions and the corresponding lactams **3ad**–**3af** were obtained in 48-73% yield. These results demonstrate that the current method provides a reliable and powerful protocol for access to functionalized lactams.

The substrates scope of the cascade reactions was further extended to 3-(2-isocyanobenzyl)-indoles (Fig. 4a). A variety of 3-(2-aminophenyl)-tetrahydroquinolines **5a**–**5f** were successfully synthesized via a cascade reaction between a diverse array of 3-(2-isocyanobenzyl)-indoles and diphenylphosphine oxide **2a**. The reaction was highly compatible with 3-(2-isocyanobenzyl)-indole from Clofibrate, yielding the desired tetrahydroquinoline **5g** with a 69% yield. Moreover, diarylphosphine oxides with different groups demonstrated high efficacy in the reaction, giving tetrahydroquinolines **5h** or **5i** with high yields. To further demonstrate the versatility of this catalytic platform, the scope was expanded to include additional radical precursors (Supplementary Fig. 19). Notably, a diverse set of thiophenols **6a**–**6e** proved competent after optimization, delivering the corresponding spiroindolenines **7a**–**7e** in moderate yields (Fig. 4b).

### Practicality of the reactions
To further demonstrate the practicality of this reaction, the cascade reaction between **1a** and **2a** was carried out on a gram scale under three blue LED lamps and 1200 rpm. The corresponding product **3a**, could be obtained in a 70% yield (Fig. 5a). To assess the synthetic utility of this methodology, lactam **3a** was subjected to a reduction with LiAlH$_4$, delivering the lactam **8** with a yield of 56% (Fig. 5b). Moreover, the Boc-protected Kynuramine **9**, identified as a potential inhibitor of neural nitric oxide synthase, was synthesized with a yield of 45% via a deprotection of the Boc-protected precursor **3a** using lithium hydroxide (Fig. 5b). Functionalized lactam **11**, which shares a structural

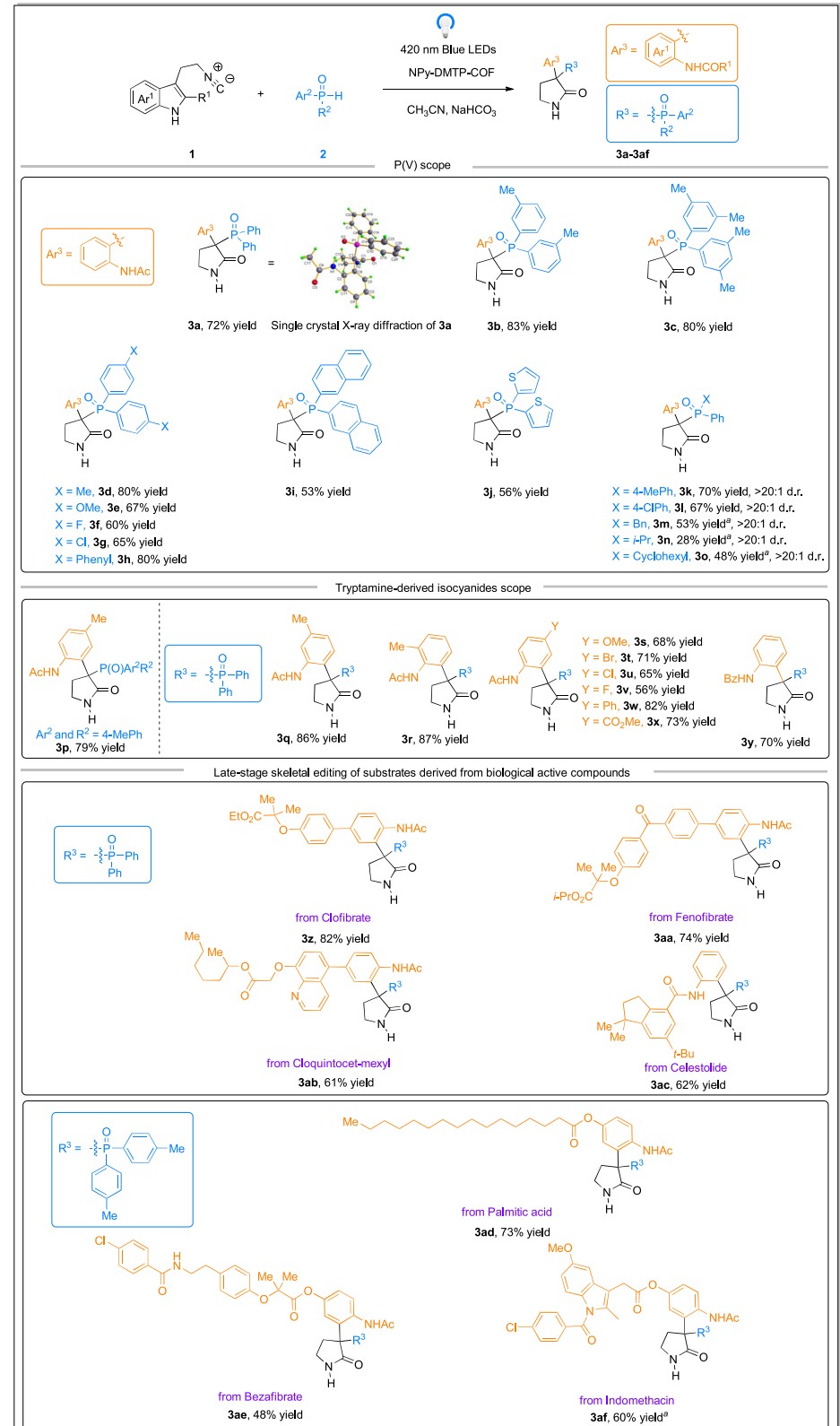

**Fig. 3 | Substrates scope of photocatalytic reactions between tryptamine-derived isocyanides and phosphine oxides.** All reactions were carried out on a 0.3 mmol scale with 1.0 eq. isocyanides **1**, 2.0 eq. phosphine oxides **2**, 2 mol% NPy-DMTP-COF and 2.0 eq. NaHCO$_3$ in 4.0 mL CH$_3$CN under irradiation of 420 nm blue LEDs with N$_2$ protection at room temperature for 48–96 h. Isolated yield. [a]2.0 eq. Na$_2$HPO$_4$ served as base instead of NaHCO$_3$.

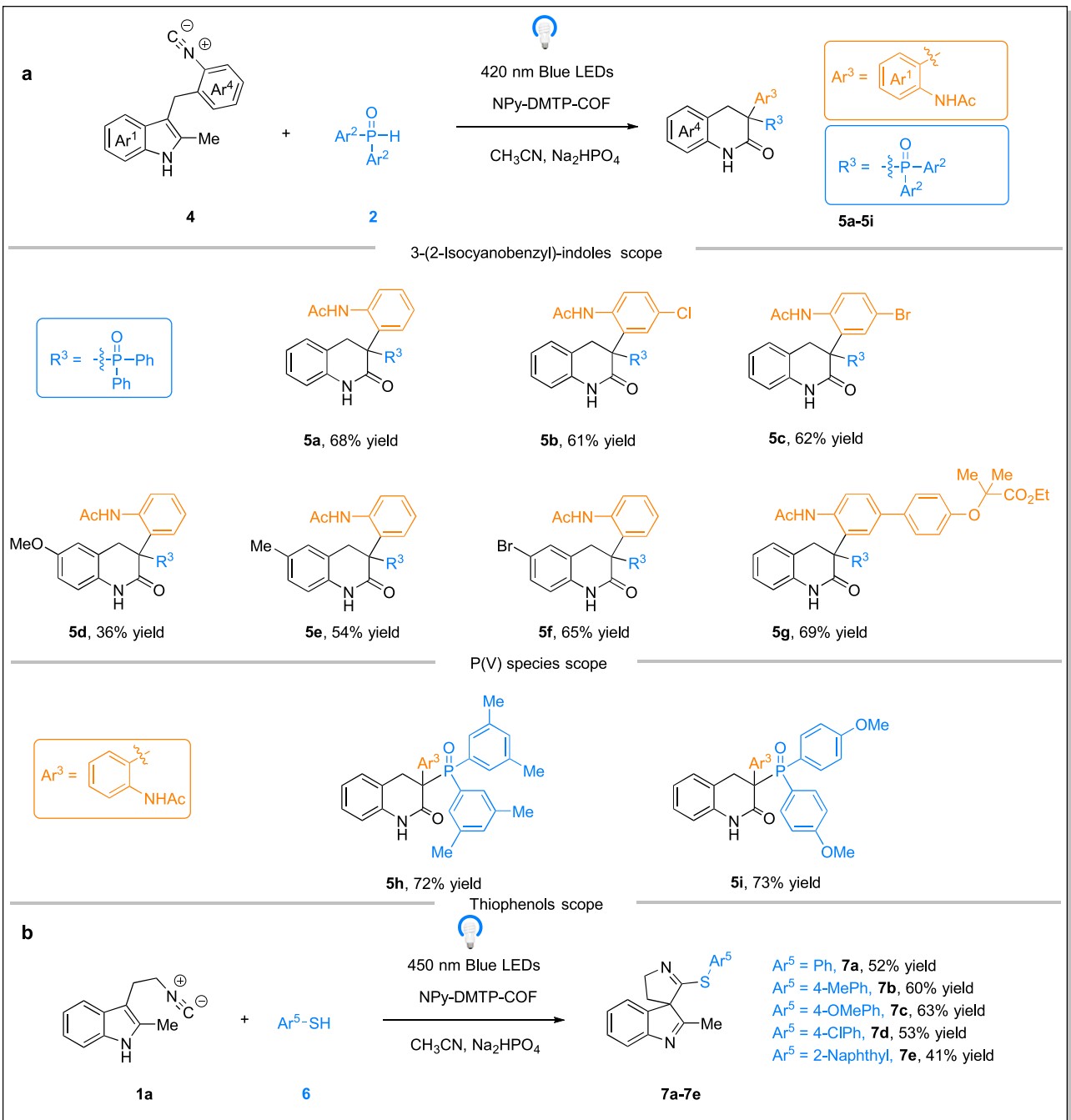

**Fig. 4 | Substrate scope for (a) photocatalytic reactions of 3-(2-isocyanobenzyl)-indoles with phosphine oxides and (b) photocatalytic cyclizations of tryptamine-derived isocyanides with thiophenols. a** All reactions were carried out on a 0.3 mmol scale with 1.0 eq. isocyanides **4**, 2.0 eq. phosphine oxide**s 2**, 2 mol% NPy-DMTP-COF and 2.0 eq. Na$_2$HPO$_4$ in 4.0 mL CH$_3$CN under irradiation of 420 nm blue LEDs with N$_2$ protection at room temperature for 48–96 h. **b** All reactions were carried out on a 0.3 mmol scale with 1.0 eq. isocyanides 1, 2.0 eq. thiophenols 6, 2 mol% NPy-DMTP-COF and 2.0 eq. Na$_2$HPO$_4$ in 4.0 mL CH$_3$CN under irradiation of 450 nm blue LEDs with N$_2$ protection at room temperature for 48–96 h. Isolated yield.

resemblance with Chimonamidine, was successfully synthesized through the reduction of the *N*-methyl protected intermediate **3a** using DIBAL (Fig. 5c). Finally, the heterogeneous nature of COF endowed it with recyclability and reusability. After the completion of the reaction, NPy-DMTP-COF could be readily recovered by simple centrifugation. In the reaction between **1a** and **2a**, NPy-DMTP-COF was successfully retrieved and reused across four subsequent cycles, with little diminution in its catalytic performance (Fig. 5d). PXRD analysis of the recycled NPy-DMTP-COF displays only a slight decline in crystallinity (Supplementary Fig. 20), while BET measurements confirm the preservation of its porous architecture, with a surface area of 1214 m$^2$/g,

pore volume of 0.85 cm$^3$/g, and pore width of 2.27 nm (Supplementary Fig. 21, Supplementary Fig. 22 and Supplementary Table 3).

## Mechanistic studies

To gain insight into the reaction mechanism, a series of mechanistic studies were carried out (Fig. 6). Initially, 2.0 equiv of 2,2,6,6-tetra-methylpiperidin-1-yl)-oxidanyl (TEMPO) was enough to quench the reaction of **1a** and **2a** (Fig. 6a). A TEMPO-**2a** adduct was detected by high-resolution mass spectrometry, indicating that phosphoryl radical might be generated photochemically (Supplementary Fig. 23). Furthermore, when the reaction between **1a** and **2a** was carried out under

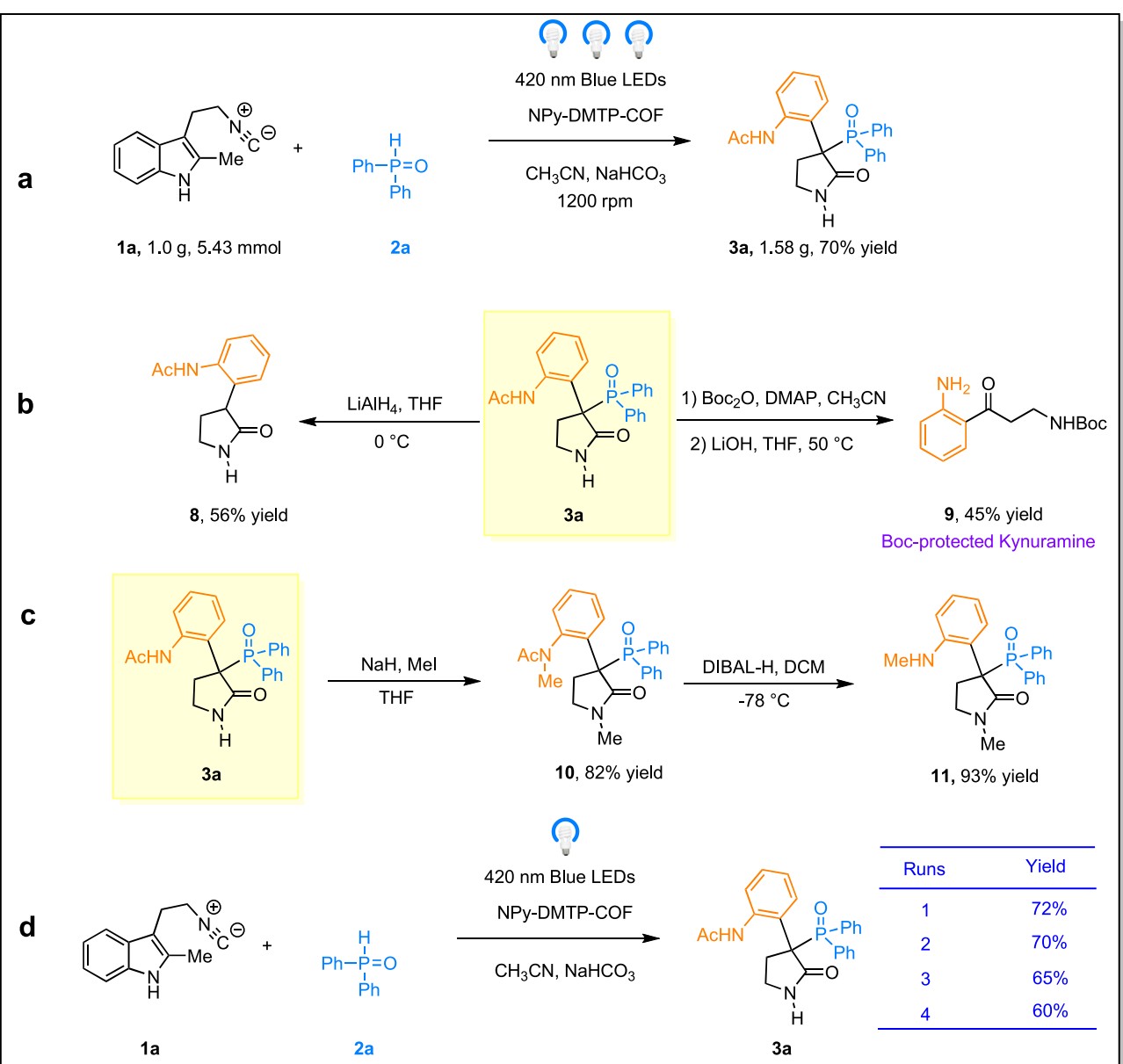

**Fig. 5 | Practicality of the reactions. a** Gram scale reaction. **b, c** Synthetic transformation. **d** Yields of **3a** with recovered NPy-DMTP-COF in four consecutive runs.

optimal conditions for 12 h, the spiroindolenine **12** was isolated with a yield of 32% (Fig. 6b) and $H_2$ was detected by gas chromatography (GC) (Supplementary Fig. 24). Spiroindolenine **12** was then employed as a substrate in a photocatalytic reaction, successfully yielding lactam **3a** with a 75% yield, and $H_2$ was again identified by GC analysis (Supplementary Fig. 25). The reaction could not proceed in the absence of base, photocatalyst or blue LEDs. These experiments indicate that spiroindolenine **12** is the key intermediate in this cascade reaction. Moreover, it suggests that the $HCO_3^-$ may act as an electron acceptor, abstracting an electron from COF⁻ to regenerate the ground state COF, while simultaneously producing $H_2$ as a byproduct. Subsequently, to elucidate the interaction between NPy-DMTP-COF and the phosphine oxide, fluorescence quenching experiments (Supplementary Figs. 26-28) and Stern-Volmer analysis (Fig. 6c) were performed. The luminescence emission of NPy-DMTP-COF was quenched effectively by [**2a** + NaHCO₃] mixture instead of **2a**, indicating that an interaction between NPy-DMTP-COF and [**2a** + NaHCO₃] mixture might exist to promote this reaction. The results above hint at an exclusive PCET process between NPy-DMTP-COF and phosphine oxide.

Moreover, an isotope-labeling study was carried out using deuterated D-P(O)Ph₂ and isocyanide **1a** under standard reaction conditions, leading to the synthesis of **3a** in a 69% yield with no hydrogen atom being deuterated (Fig. 6d, Supplementary Fig. 29 and Supplementary Fig. 30). It suggests that the deuterium atom in D-P(O)Ph₂ is likely captured by NaHCO₃.

To explore the potential pathways involved in the photocatalytic cascade reaction, we employed DFT calculations, incorporating Grimme's D3 dispersion correction (Fig. 7). All calculations were carried out with the Gaussian16 program package[93]. All the structures were fully optimized based on PBE0-D3/6-311 G(d,p) basis set level[94,95]. All optimized geometries were characterized as either local minima (no imaginary frequencies) or transition states (one imaginary frequency) through vibrational frequency analysis at the same level of theory. The connection of each transition state was validated by intrinsic reaction coordinate (IRC) calculations. Subsequently, Gibbs free energies were computed via M06-2X-D3/def2TZVP single-point energy calculations on the optimized structures, incorporating thermal corrections at 298.15 K and 1.0 atm. TD-DFT study was conducted

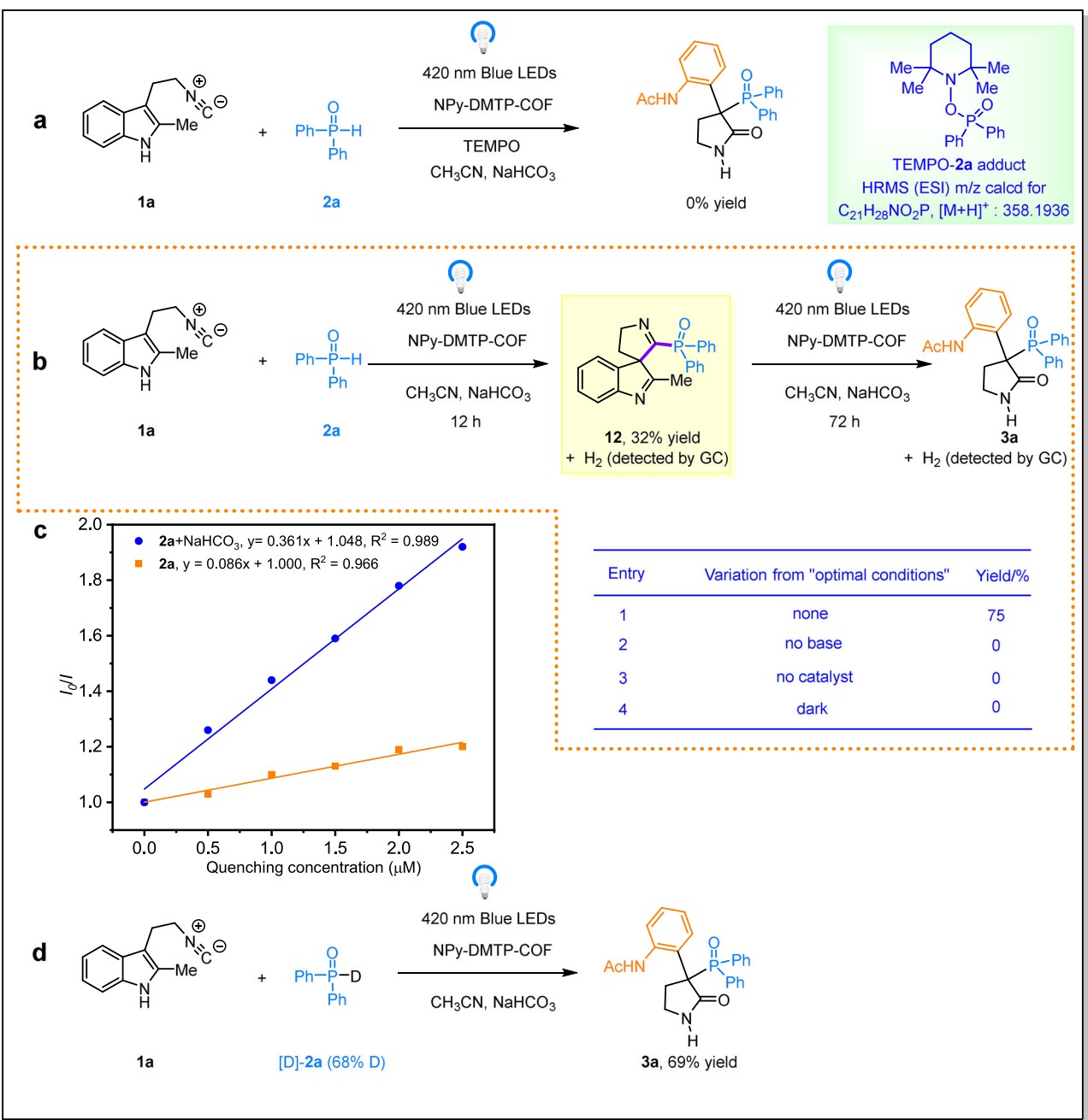

**Fig. 6 | Controlled experiments. a** Radical prevention experiment. **b** The experiment involving in the isolation of intermediate **12** and the synthesis of **3a** from **12** via photocatalytic reaction. **c** Stern-Volmer quenching studies. **d** Deuterium labeling experiment.

at same level within the adiabatic approximation to predict the excitation energies. The Marcus theory was applied to evaluate the PCET process and the single electron transfer process. To evaluate solvation effects of acetonitrile, the IEFPCM model was applied based on the gas-phase optimized structures of all the species.

Initially, NPy-DMTP-COF is selectively excited under blue LEDs, with a free energy excitation of 64.3 kcal/mol, which is significantly lower than that of the reactants **1a** and **2a**, at 126.3 kcal/mol and 145.1 kcal/mol, respectively. This initiates a PCET process, where an electron is transferred from **2a** to NPy-DMTP-COF, accompanied by a proton transfer from **2a** to $HCO_3^-$. This results in the formation of a phosphoryl radical and a COF radical anion, NPy-DMTP-COF·⁻. The phosphoryl radical then reacts with **1a** to form **Int-1**, with a Gibbs free energy change of −0.9 kcal/mol. Potential energy surface (PES) scanning confirms that this reaction step is barrier-free. **Int-1** subsequently

undergoes cyclization to form **Int-2** via the transition state **Ts-1**, requiring an activation energy of 10.5 kcal/mol. **Int-2** undergoes an electron transfer with NPy-DMTP-COF⁺ and $HCO_3^-$ to yield spiroindolenine **12**, a spontaneous process with a Gibbs free energy change of 14.4 kcal/mol (Fig. 7a). The formation of spiroindolenine **12** has been confirmed experimentally (Fig. 6b).

A nucleophilic addition then occurs, where a hydroxide anion attacks spiroindolenine **12**, followed by semipinacol rearrangement and tautomerism to yield **Int-7**. This reaction pathway splits into two routes. The favored route (black line) involves the formation of **Int-3**, followed by a ring-opening reaction via **Ts-2**, with an energy barrier of 5.1 kcal/mol, leading to **Int-5**. The subsequent rate-determining step involves 1,2-phosphonyl migration through **Ts-4**, with an energy barrier of 20.7 kcal/mol, leading to the formation of **Int-6**. Tautomerization through **Ts-5**, with an energy barrier of 17.6 kcal/mol yields amide

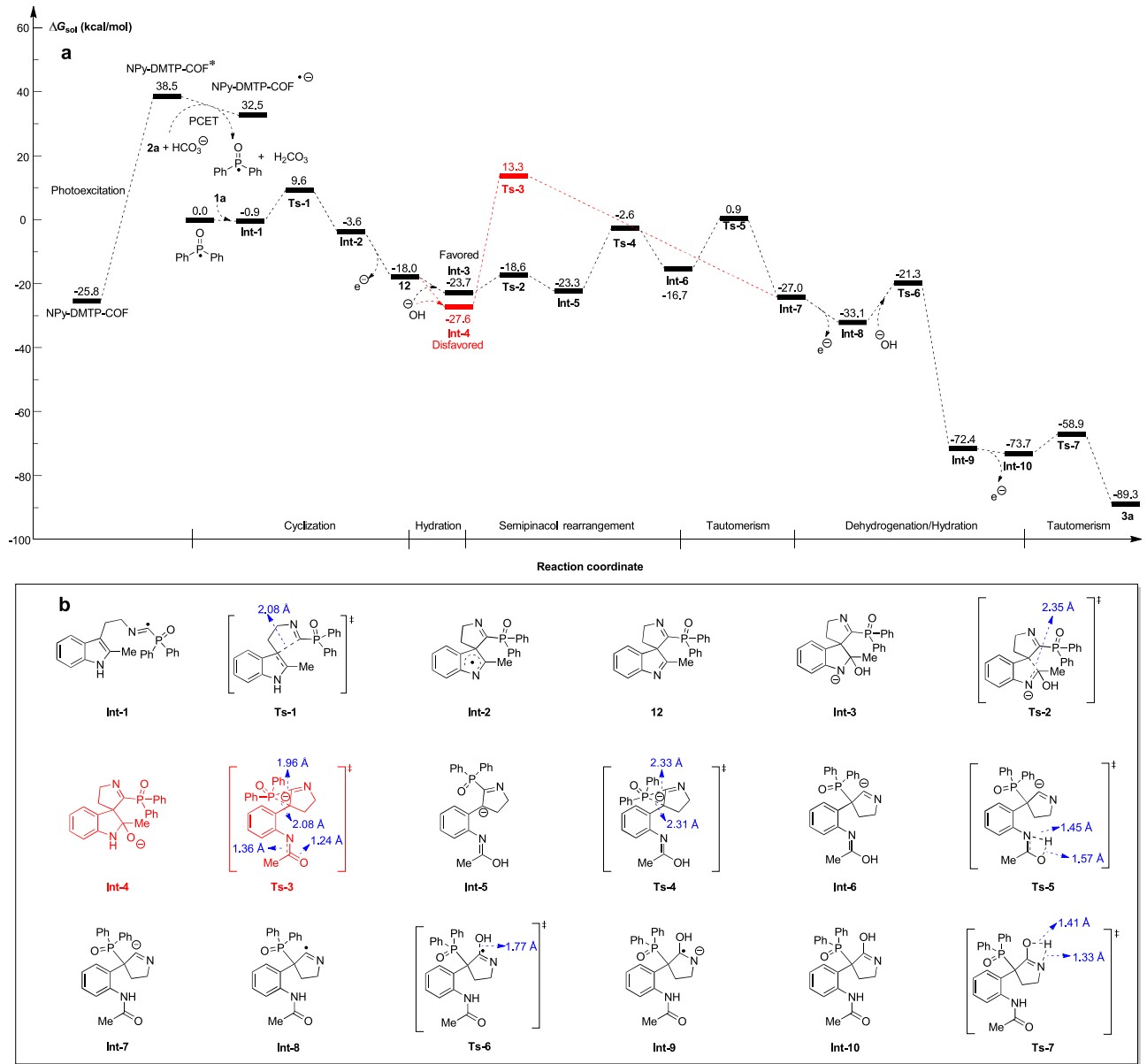

**Fig. 7 | The DFT studies of photocatalytic cascade reaction between 1a and 2a.** All energies were calculated at the M06-2X-D3 functional level, utilizing the def2-TZVP basis set. **a** Gibbs free energy of reaction paths. **b** Intermediates and transition states for this catalytic system.

**Int-7**. The alternative route (red line), involving a one-step rearrangement from **Int-4** to **Ts-3**, is less favored due to a high energy barrier of 40.9 kcal/mol.

**Int-7** undergoes a SET process to generate the radical intermediate **Int-8**. Nucleophilic attack by a hydroxide anion at the radical center then proceeds via **Ts-6** with an energy barrier of 11.8 kcal/mol to afford the stabilized anionic species **Int-9**. A subsequent SET process from **Int-9** yields the neutral intermediate **Int-10**, which tautomerizes through **Ts-7** with an energy barrier of 14.8 kcal/mol to deliver the final product **3a**. All the intermediates and transition states for this catalytic system are illustrated in Fig. 7b.

Here, we report the construction of a well-ordered framework, NPy-DMTP-COF, with designed porosity that can be efficiently excited under visible light. This catalyst facilitates a photocatalytic cascade reaction between tryptamine-derived isocyanides and phosphine oxides, yielding a diverse array of 3-(2-aminophenyl)-γ-lactams with moderate to high yields and excellent diastereoselectivities (> 20:1). 3-(2-isocyanobenzyl)-indoles are also tolerated, giving (2-aminophenyl)-tetrahydroquinolines in moderate yield. Notably, NPy-DMTP-COF could be readily recovered via simple centrifugation and reused over four consecutive cycles without noticeable loss of catalytic activity, highlighting its advantage over conventional homogeneous catalysts. The controlled experiments and DFT studies reveal that a PCET process occurs between the excited NPy-DMTP-COF* and phosphine oxide, initiating a cascade reaction including cyclization, semipinacol rearrangement and dehydrogenation/hydration processes. Notably, the in situ generated spiroindolenine serves as the key intermediate in this reaction. The current work not only expands the repertoire of 3-(2-aminophenyl)-γ-lactams and (2-aminophenyl)-tetrahydroquinolines but also establishes a COF-based photocatalytic system for cascade reactions.

## Methods

### General information

Reactions were monitored by thin layer chromatography (TLC), and column chromatography purifications were carried out using silica gel.

Column chromatography was performed on silica gel (300–400 mesh). $^1$H, $^{13}$C, $^{19}$F and $^{31}$P NMR spectra were collected on a Bruker AV 400, 600 or 700 MHz NMR spectrometer using residue solvent peaks as an internal standard ($^1$H NMR: CDCl$_3$ at 7.26 ppm, DMSO-$d_6$ at 2.50 ppm; $^{13}$C NMR: CDCl$_3$ at 77.00 ppm, DMSO-$d_6$ at 30.92 ppm). Data for $^1$H and $^{13}$C NMR were recorded as follows: chemical shift (δ, ppm), multiplicity (s = singlet; d = doublet; t = triplet; q = quarter; m = multiplet; br = broad), coupling constant (Hz), integration. High resolution mass spectra for all the new compounds were done by an LTQ-Orbitrap instrument (ESI) (Thermo Fisher Scientific, USA). Catalysts and substrates **2** and **6** were purchased from J&K Scientific (China) Co., LTD. Substrates **1** and **4** were synthesized by following the published procedures[35].

Powder X-ray powder diffraction (PXRD) patterns were measured on a Rigaku RINT Smartlab SE powder diffraction system, equipped with Cu Kα radiation (λ = 1.54 Å). Fourier-transform infrared spectrometer (FT-IR) data were collected using a Bruker VERTEX 70 spectrophotometer with KBr disks. Solid-state $^{13}$C-CP/TOSS NMR spectra were performed on a Bruker 500 MHz spectrometer. Nitrogen sorption isotherms were recorded on a Quantachrome Autosorb 1 at 77 K. Prior to the measurements, the samples were outgassed for 12 h at 120 °C under high vacuum. Scanning electron microscopy (SEM) was performed using a ZEISS Gemini 560 operated at an accelerating voltage of 2.0 kV. High resolution transmission electron microscopy (HR-TEM) was performed using a JEM-2100F instrument (JEOL, Japan) operated at 200 kV. Dynamic light scattering (DLS) was performed on Zeta-potential & Particle size analyzer (Malvern ZS90). NPy-DMTP-COF was dispersed in CH$_3$CN (0.05 mg/mL), sonicated for 60 min, and left to stand for 10 min prior to DLS measurements. Thermal gravimetric analyses (TGA) were performed on a Mettler Toledo's TGA/DSC 3+ Synchronous Thermal Analyzer. The process of simulating COF structure was performed by the Materials Studio software. The triclinic lattice with P1 symmetry group was set as the initial eclipsed COF structure. After the smallest asymmetric fragment was filled into the blank cell, the Forcite tools package was employed to optimize the cell geometry including energy minimization. The cell optimized from the Universal force fields was subsequently refined using the Pawley refinement method in Reflex tools.

### Preparation of NPy-DMTP-COF

6,6′,6″,6‴-(pyrene-1,3,6,8-tetrayl)tetrakis(naphthalen-2-amine) (NPy) (38.3 mg, 0.05 mmol) and 2,5-dimethoxyterephthalaldehyde (DMTP) (19.4 mg, 0.1 mmol) and CH$_3$COOH (0.1 mL, 6 M) were mixed in a mixture of solvents containing 0.25 mL of o-dichlorobenzene and 0.75 mL of 1-butanol in a sealed tube. The mixture was ultrasonicated for 10 min and then flash frozen at 77 K (liquid N$_2$ bath) and degassed through three freeze-pump-thaw cycles. The mixture was then heated at 120 °C for 72 h under an N$_2$ atmosphere. The resulting solid product was recovered by filtration, washed with methanol and DCM, and subjected to Soxhlet extraction using THF for 24 h. After drying under vacuum for 6 h, a brown solid was obtained with a yield of 87%. This material was designated as NPy-DMTP-COF.

### General experimental procedure of photocatalytic cascade reaction

At N$_2$ atmosphere, NPy-DMTP-COF (5.3 mg, 0.006 mmol based on the repeating unit), **1** or **4** (0.3 mmol), **2** (0.6 mmol), and NaHCO$_3$ (50.4 mg, 0.6 mmol) were mixed in acetonitrile (4.0 mL) in a Schlenk tube. The resulting mixture was stirred under blue LED irradiation (420 nm) at room temperature for 48–96 h. Upon completion, the reaction mixture was filtered through celite pad and the filtrate was concentrated. The mixture was purified by silica gel flash chromatography (petroleum ether:ethyl acetate, 1:3) to afford the desired product **3** or **5**.

## Data availability

Crystallographic data for the structures reported in this article have been deposited at the Cambridge Crystallographic Data Center, under deposition numbers CCDC 2368727 (**3a**). Copies of the data can be obtained free of charge via https://www.ccdc.cam.ac.uk/structures/. The characterizations of NPy-DMTP-COF, spectral data of products and cartesian coordinates generated in this study are provided in the Supplementary Information and Source Data file. All data are available from the corresponding author upon request. Source data are provided with this paper.

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

## Acknowledgements

This research was supported by the Department of Science and Technology of Fujian Province (No.2022H6021) from Z.Y., the Forestry Science and Technology Project of Fujian Province (No. 2023FKJ27) from Z.Y. We also acknowledge financial support from the JST-ERATO Yamauchi Materials Space-Tectonics Project (JPMJER2003) and the ARC Laureate Fellowship (FL230100095) from Y.Y. and the ARC Discovery Early Career Researcher Award (DECRA) (DE220101577) from J.W. This work used the Queensland node of the NCRIS-enabled Australian National Fabrication Facility (ANFF) from Y.Y. We demonstrate our appreciation for English editing software such as ChatGPT, Grammarly, etc.

## Author contributions

X.L. conceived the idea, executed most experiments and wrote the article. J.L. and J.Z. revised the article. X.C. synthesized several substrates. R.S. conducted the DFT studies. D.J. performed HR-TEM, BET measurements and revised the article. Y.K. and J.W. performed the SEM and BET measurements. L.W.; C.S.; Y.Y., and Z.Y. supervised the research project. All authors discussed the results and were engaged in revising and completing the final version of the manuscript.

## Competing interests

The authors declare no competing interests.
