## [Transparent Peer review file · Nature Communications]

Straightforward Construction of Functionalized γ -Lactams via Conjugated-engineered Covalent Organic Framework Photocatalysed Cascade Reactions

Corresponding Author: Professor Zhanhui Yuan

Version 0:

Reviewer comments:

Reviewer #1

(Remarks to the Author)

This research work successfully accomplishes visible-light-induced catalytic cascade reactions between tryptamine-derived isocyanides and phosphine oxides, with NPy-DMTP-COF serving as the catalyst. A diverse range of 3-(2-aminophenyl)- γ -lactams are synthesized with good yields under mild reaction conditions. Moreover, 3-(2-isocyanobenzyl)-indoles are also found to be compatible with this catalytic system, enabling the formation of (2-aminophenyl)-tetrahydroquinolines in moderate yields. Both experimental investigations and theoretical calculations clearly indicate that a proton-coupled electron transfer (PCET) process occurs between the excited state of NPy-DMTP-COF*, i.e., NPy-DMTP-COF in its excited state, and the phosphine oxide. This PCET event triggers a series of subsequent cascade reactions, encompassing cyclization, semipinacol rearrangement, and dehydrogenation/hydration processes.

Achieving the synthesis of functionalized γ -lactams through a cascade radical process via combined heterogeneous catalysis remains a formidable challenge. This reviewer is highly enthusiastic about recommending the publication of this work in this journal, provided that some minor revisions are made.

(1) In Table 1, the author has chosen several homogeneous catalysts for this catalytic system. However, it would be interesting to explore the outcomes when using certain heterogeneous catalysts, such as g-C₃N₄ and graphene oxide.

(2) It would be more advantageous to merge the two parts presented in Figure 7 into a single image.

Reviewer #2

(Remarks to the Author)

Manuscript ID: NCOMMS-25-26352-T

Title: Straightforward Construction of Functionalized γ -Lactams via Conjugated-engineered Covalent Organic Framework Photocatalysed Cascade Reactions

Author(s): Xiangfeng Lin, Jianguo Li, Jiaxian Zheng, Xiaowei Cai, Liwei Wang, Rongjian Sa, Chuanling Si, Dong Jiang, Yusuke Yamauchi, Zhanhui Yuan

General Comments:

This manuscript presents a novel photocatalytic cascade reaction for synthesizing functionalized γ -lactams using a naphthyl-based pyrene-containing covalent organic framework (NPy-DMTP-COF) as a heterogeneous photocatalyst. The work is well-executed, with comprehensive characterization of the COF, detailed mechanistic studies, and broad substrate scope exploration. The study addresses an important synthetic challenge and demonstrates the potential of COFs in visible-light-driven catalysis. This work represents a significant advance in COF-photocatalyzed synthesis, offering a sustainable route to γ -lactams with high diastereoselectivity. Addressing the major points—particularly mechanistic validation and catalyst stability—would elevate the manuscript's impact. With revisions, this study would be a strong candidate for publication in a high-impact journal.

Specific comments:

1) The PXRD and BET analyses confirm the crystallinity and porosity of NPy-DMTP-COF and this COF can be recycled (4 cycles) without losing reactivity. The manuscript should address whether the COF's crystallinity and porosity are retained

after reuse.

- 2) The scope focuses heavily on diarylphosphine oxides. Extending to alkylphosphine oxides or other radical precursors (e.g., boronic acids) could demonstrate broader utility. Could the authors try several other radical precursors?
- 3) The superiority of NPy-DMTP-COF over Py-DMTP-COF is attributed to conjugation effects, but a side-by-side comparison of their optical/electrochemical properties (e.g., UV-vis, CV) would clarify this rationale.
- 4) The DFT energy diagrams (Fig. 7) are incomprehensible. However, the Gibbs free energy of reaction paths from 10 to 3a is misleading. As shown in Fig. 7b, only two TS were found for the main path. Moreover, the formation of Int-6, P-shift via Ts-4, with an energy barrier of 28.2 kcal/mol, is theoretically unreliable. Furthermore, the Cartesian Coordinates are not provided in the Supplementary Information. Therefore, the DFT calculations require more thorough explanations.
- 5) Fluorescence quenching experiments provide critical insights into molecular interactions, dynamic processes, and material properties by analyzing how and why a fluorophore's emission is suppressed. Therefore, fluorescence quenching experiments are suggested to be carried out.
- 6) Some of the spectra contain impurities, for examples: 3n, 3m

Reviewer #3

(Remarks to the Author)

Lin et al. describe the use of a COF-based photocatalyst to facilitate cascade reactions for the synthesis of functionalized γ -lactams. The study is well-conceived, experimentally thorough, and clearly presented. However, I suggest the authors address the following points before the manuscript is recommended for acceptance:

- 1) Thermogravimetric Analysis (TGA): The TGA plot in Figure S8 exhibits unusual features. Specifically, there appears to be an increase in mass between 150 and 400 °C, which is not typical in decomposition profiles. Could the authors clarify this behavior? Additionally, the small weight gain or step around 500 °C might indicate oxidative processes or residual metal content—any insights would be appreciated.
- 2) Particle Size Information: The manuscript does not report the particle size of the as-synthesized COF. Given the solid-state nature of the catalyst, this information is essential for understanding its dispersion, light absorption efficiency, and catalytic behavior. Please provide size characterization data (e.g., SEM/TEM, DLS).
- 3) Scalability and Light Penetration Issues: The drop in yield from 73% (small scale) to 43% (gram scale) could stem from limited light penetration in heterogeneous systems. Could the authors explore whether reducing the particle size or optimizing stirring conditions improves the yield at larger scales?
- 4) Necessity of Heterogeneous Catalysis: To evaluate whether the extended framework of the COF is essential for photocatalysis, I recommend comparing its performance to a discrete molecular analogue. For example, a Schiff base compound synthesized from 1,3,6,8-tetrakis(2-aminonaphthyl)pyrene and 2-methoxybenzaldehyde could serve as a molecular mimic of the NPy-DMTP-COF. Such a control would provide valuable insights into the role of the COF structure in the observed reactivity.

Version 1:

Reviewer comments:

Reviewer #1

(Remarks to the Author)

In this revised manuscript, the authors have tested both commercial g-C₃N₄ and graphene oxide under identical conditions, and have revised the DFT calculations and Figure 7 as appropriate. Having carefully reviewed the revised manuscript, I am pleased to report that the authors have thoroughly addressed all my suggestions, along with those from other reviewers. The revised version now meets the standards required for publication in Nature Communications. I am satisfied with its current format and consider it now acceptable for publication.

Reviewer #2

(Remarks to the Author)

Lin et al. have carefully revised the manuscript by adding more examples and providing more comments for better understandings. As replied, all the concerns, including substrate scope, recyclability, mechanistic study, and theoretical calculation have been addressed. Therefore, this reviewer agrees that it is suitable for the acceptance of Nature Communications journal.

Reviewer #3

(Remarks to the Author)

The authors have adequately addressed my previous concerns, and I am happy to recommend the manuscript for publication.

Reviewer #1 (Remarks to the Author):

This research work successfully accomplishes visible-light-induced catalytic cascade reactions between tryptamine-derived isocyanides and phosphine oxides, with **NPy-DMTP-COF** serving as the catalyst. A diverse range of 3-(2-aminophenyl)- γ -lactams are synthesized with good yields under mild reaction conditions. Moreover, 3-(2-isocyanobenzyl)-indoles are also found to be compatible with this catalytic system, enabling the formation of (2-aminophenyl)-tetrahydroquinolines in moderate yields. Both experimental investigations and theoretical calculations clearly indicate that a proton-coupled electron transfer (PCET) process occurs between the excited state of **NPy-DMTP-COF***, i.e., **NPy-DMTP-COF** in its excited state, and the phosphine oxide. This PCET event triggers a series of subsequent cascade reactions, encompassing cyclization, semipinacol rearrangement, and dehydrogenation/hydration processes. Achieving the synthesis of functionalized γ -lactams through a cascade radical process via combined heterogeneous catalysis remains a formidable challenge. This reviewer is highly enthusiastic about recommending the publication of this work in this journal, provided that some minor revisions are made.

Q1. In Table 1, the author has chosen several homogeneous catalysts for this catalytic system. However, it would be interesting to explore the outcomes when using certain heterogeneous catalysts, such as g-C₃N₄ and graphene oxide.

A1. Thank you for your suggestions. We examined commercial g-C₃N₄ and graphene oxide under identical conditions, yet no product was detected (Table 1, entries 8 and 9), probably owing to their absence of well-defined porosity and extended π -conjugation.

Q2. It would be more advantageous to merge the two parts presented in Figure 7 into a single image.

A2. Thank you very much for your valuable suggestion. We have carefully revised **Figure 7** by merging the two parts into a single, integrated image to improve clarity and readability.

Reviewer #2 (Remarks to the Author):

This manuscript presents a novel photocatalytic cascade reaction for synthesizing functionalized γ -lactams using a naphthyl-based pyrene-containing covalent organic framework (**NPy-DMTP-COF**) as a heterogeneous photocatalyst. The work is well-executed, with comprehensive characterization of the COF, detailed mechanistic studies, and broad substrate scope exploration. The study addresses an important synthetic challenge and demonstrates the potential of COFs in visible-light-driven catalysis. This work represents a significant advance in COF-photocatalyzed synthesis, offering a sustainable route to γ -lactams with high diastereoselectivity. Addressing the major points, particularly mechanistic validation and catalyst stability, would elevate the manuscript's impact. With revisions, this study would be a strong candidate for publication in a high-impact journal.

Specific comments:

Q1. The PXRD and BET analyses confirm the crystallinity and porosity of **NPy-DMTP-COF** and

this COF can be recycled (4 cycles) without losing reactivity. The manuscript should address whether the COF's crystallinity and porosity are retained after reuse.

A1. Thank you very much for your constructive suggestions. We have performed PXRD and BET analyses on the recycled **NPY-DMTP-COF** after photocatalysis. The PXRD pattern of the reused catalyst demonstrates that it retains excellent crystallinity after photocatalysis (Supplementary Fig. 20), while BET measurements confirmed that its porous architecture is largely retained, which the surface area is 1214 m²/g, the pore volume of 0.85 cm³/g, and pore width of 2.27 nm (Supplementary Fig. 21, Supplementary Fig. 22 and Supplementary Table 3). These results aligned with the little deterioration in photocatalytic activity.

Q2. The scope focuses heavily on diarylphosphine oxides. Extending to alkylphosphine oxides or other radical precursors (e.g., boronic acids) could demonstrate broader utility. Could the authors try several other radical precursors?

A2. Thank you very much for the valuable suggestions. To evaluate the broader applicability of our protocol, we employed phenylalkylphosphine oxides as phosphorus sources, which were found to be compatible with the reaction conditions and furnished products **3m–3o** in moderate yields with excellent diastereoselectivities (d.r. > 20 : 1). We also made deliberate attempts to extend the scope to dialkylphosphine oxides under identical conditions, however, no conversion of **1a** was observed, presumably due to the disfavoured PCET process between dialkylphosphine oxides and **NPY-DMTP-COF***. In addition, we explored a variety of other radical precursors. Notably, a diverse set of thiophenols **6a–6e** proved competent after optimization, affording the corresponding spiroindolenines **7a–7e** in moderate yields (Fig. 4b). Other tested radical precursors that were unsuccessful are summarized in the Supplementary Information (Supplementary Fig. 19)

Q3. The superiority of **NPY-DMTP-COF** over **Py-DMTP-COF** is attributed to conjugation effects, but a side-by-side comparison of their optical/electrochemical properties (e.g., UV-vis, CV) would clarify this rationale.

A3. Thank you very much for your insightful suggestion. To clarify this point, we have provided a side-by-side comparison of the optical and electrochemical properties of **NPY-DMTP-COF** and **Py-DMTP-COF** (Supplementary Fig. 11 and Supplementary Fig. 14). The UV-vis spectra indicates that both photocatalysts exhibit very similar absorption profiles at around 420 nm (Supplementary Fig. 11). Therefore, the underlying rationale behind this disparity might be attributed to the redox potential of photocatalysts. Cyclic voltammetry measurements reveal that **NPY-DMTP-COF** exhibits a higher redox potential of -0.72 V (PC/PC⁻) compared with -1.05 V for **Py-DMTP-COF** (Supplementary Fig. 14). As a result, the excited-state **NPY-DMTP-COF*/NPY-DMTP-COF⁻*** potential is significantly more positive, which suggests that the excited **NPY-DMTP-COF** is more capable of abstracting an electron from the P-radical precursor, thereby facilitating the PCET process.

Q4. The DFT energy diagrams (Fig. 7) are incomprehensible. However, the Gibbs free energy of reaction paths from **10** to **3a** is misleading. As shown in Fig. 7b, only two TS were found for the main path. Moreover, the formation of Int-6, P-shift via Ts-4, with an energy barrier of 28.2 kcal/mol, is theoretically unreliable. Furthermore, the Cartesian Coordinates are not provided in

the Supplementary Information. Therefore, the DFT calculations require more thorough explanations.

A4. Thank you very much for this valuable suggestion. We have thoroughly revised the DFT energy diagram for the conversion from **12** to **3a** (Fig. 7) to present a more comprehensive and theoretically reliable mechanistic picture. In the updated pathway, five distinct transition states have now been identified, addressing a key limitation of the previous version, which included only two. The transition state **Ts-4**, corresponding to the 1,2-phosphonyl shift leading to **Int-6**, has been carefully reevaluated using a tighter integration grid and solvent stabilization. The originally reported energy barrier of 28.2 kcal·mol⁻¹ appeared unusually high, and re-optimization under improved computational settings yielded a corrected barrier of 20.7 kcal·mol⁻¹. This revised value aligns more closely with established rearrangement energetics, rendering the proposed pathway mechanistically more plausible under mild conditions. Moreover, all Cartesian coordinates for the optimized molecules, intermediates, and transition states are now provided in the Supplementary Information.

Q5. Fluorescence quenching experiments provide critical insights into molecular interactions, dynamic processes, and material properties by analyzing how and why a fluorophore's emission is suppressed. Therefore, fluorescence quenching experiments are suggested to be carried out.

A5. Thank you very much for this valuable suggestion. We performed fluorescence quenching experiments (Supplementary Figs. 26-28) and conducted Stern-Volmer analysis (Fig. 6c) to gain further insights into the interaction between **NPy-DMTP-COF** and the phosphine oxide. The luminescence emission of **NPy-DMTP-COF** was effectively quenched by [**2a**+NaHCO₃] mixture rather than by **2a** alone, indicating that an interaction between **NPy-DMTP-COF** and the [**2a**+NaHCO₃] mixture likely plays a critical role in promoting the reaction. These results further support the involvement of an exclusive PCET process between **NPy-DMTP-COF** and phosphine oxide.

Q6. Some of the spectra contain impurities, for examples: **3n**, **3m**.

A6. Thank you very much for this helpful suggestion. We have repeated the catalytic reactions and successfully re-purified both **3m** and **3n**, and the updated spectra have been included in the revised Supplementary Information.

Reviewer #3 (Remarks to the Author):

Lin et al. describe the use of a COF-based photocatalyst to facilitate cascade reactions for the synthesis of functionalized γ -lactams. The study is well-conceived, experimentally thorough, and clearly presented. However, I suggest the authors address the following points before the manuscript is recommended for acceptance:

Q1. Thermogravimetric Analysis (TGA): The TGA plot in Figure S8 exhibits unusual features. Specifically, there appears to be an increase in mass between 150 and 400 °C, which is not typical in decomposition profiles. Could the authors clarify this behavior? Additionally, the small weight gain or step around 500 °C might indicate oxidative processes or residual metal content—any insights would be appreciated.

A1. Thank you very much for this valuable suggestion. To clarify the unusual features observed in the initial TGA profile, we repeated the measurements using a Mettler Toledo TGA/DSC 3+ Synchronous Thermal Analyzer under carefully controlled conditions in both flowing nitrogen (Supplementary Fig. 10a) and oxygen (Supplementary Fig. 10b) atmospheres. In the repeated experiments, no abnormal mass increase was observed between 150 and 400 °C, and no noticeable inflection or step was detected around 500 °C. These results confirm the reliability of the updated data, and the corresponding figures and related descriptions in both the main manuscript and the Supplementary Information have been revised accordingly.

Q2. Particle Size Information: The manuscript does not report the particle size of the as-synthesized COF. Given the solid-state nature of the catalyst, this information is essential for understanding its dispersion, light absorption efficiency, and catalytic behavior. Please provide size characterization data (e.g., SEM/TEM, DLS).

A2. Thank you very much for this helpful suggestion. We have carried out detailed size characterization of **NPY-DMTP-COF** using multiple techniques. SEM images reveal that **NPY-DMTP-COF** exhibits a rod-like morphology with a uniform diameter of about 65 nm (Supplementary Fig. 3). HR-TEM images demonstrate a particle size of approximately 60 nm, which correlate well with the SEM measurements. (Supplementary Fig. 4). In addition, dynamic light scattering (DLS) measurements were performed under various dispersion conditions. When dispersed in acetonitrile at room temperature, **NPY-DMTP-COF** displays a Z-average size of 41.05 nm and a polydispersity index (PDI) of 0.115 (Supplementary Fig. 5), indicating excellent colloidal stability.

Q3. Scalability and Light Penetration Issues: The drop in yield from 73% (small scale) to 43% (gram scale) could stem from limited light penetration in heterogeneous systems. Could the authors explore whether reducing the particle size or optimizing stirring conditions improves the yield at larger scales?

A3. Thank you very much for this valuable suggestion. To address the reviewer's concern, we optimized the gram-scale photocatalytic reaction by employing three blue LED lamps and increasing the stirring rate to 1200 rpm, which successfully improved the isolated yield to 70% (1.58 g). We further attempted to optimize the synthetic protocol of **NPY-DMTP-COF** by systematically varying the solvent composition, acid ratio, and reaction time. However, under these modified conditions, the resulting COFs tended to form larger aggregates with somewhat reduced crystallinity, indicating that achieving smaller particle sizes without compromising framework order remains challenging, and we are continuing to investigate this important aspect in our ongoing studies.

Q4. Necessity of Heterogeneous Catalysis: To evaluate whether the extended framework of the COF is essential for photocatalysis, I recommend comparing its performance to a discrete molecular analogue. For example, a Schiff base compound synthesized from 1,3,6,8-tetrakis(2-aminonaphthyl)pyrene and 2-methoxybenzaldehyde could serve as a molecular mimic of the **NPY-DMTP-COF**. Such a control would provide valuable insights into the role of the COF structure in the observed reactivity.

A4. Thank you very much for this valuable suggestion. To evaluate the role of the extended

framework, we synthesized a discrete molecular analogue of **NPY-DMTP-COF** (NPY-MP) from 1,3,6,8-tetrakis(2-aminonaphthyl)pyrene and 2-methoxybenzaldehyde and tested it as a photocatalyst under identical reaction conditions. Negligible activity was observed for NPY-MP, indicating that the ordered and porous architecture of **NPY-DMTP-COF** is essential for achieving efficient photocatalytic performance (Table 1, entry 7).